# SARS-CoV-2 and COVID-19 Research Trend during the First Two Years of the Pandemic in the United Arab Emirates: A PRISMA-Compliant Bibliometric Analysis

**DOI:** 10.3390/ijerph19137753

**Published:** 2022-06-24

**Authors:** Basem Al-Omari, Tauseef Ahmad, Rami H. Al-Rifai

**Affiliations:** 1Department of Epidemiology and Population Health, College of Medicine and Health Sciences, Khalifa University, Abu Dhabi P.O. Box 127788, United Arab Emirates; basem.alomari@ku.ac.ae; 2KU Research and Data Intelligence Support Center (RDISC), Khalifa University of Science and Technology, Abu Dhabi P.O. Box 127788, United Arab Emirates; 3COVID-19 Research Epidemiology Sub-Committee of Abu Dhabi, Abu Dhabi Public Health Center, Abu Dhabi Department of Health, Abu Dhabi P.O. Box 5674, United Arab Emirates; 4Vanke School of Public Health, Tsinghua University, Beijing 100084, China; hamdard_hu@yahoo.com or; 5Department of Epidemiology and Health Statistics, School of Public Health, Southeast University, Nanjing 210096, China; 6Institute of Public Health, College of Medicine and Health Sciences, United Arab Emirate University, Al Ain P.O. Box 15551, United Arab Emirates

**Keywords:** COVID-19, SARS-CoV-2, bibliometric analysis, United Arab Emirates, UAE

## Abstract

Scientific research is an integral part of fighting the COVID-19 pandemic. This bibliometric analysis describes the COVID-19 research productivity of the United Arab Emirates (UAE)-affiliated researchers during the first two years of the pandemic, 2020 to 2022. The Web of Science Core Collection (WoSCC) database was utilized to retrieve publications related to COVID-19 published by UAE-affiliated researcher(s). A total of 1008 publications met the inclusion criteria and were included in this bibliometric analysis. The most studied broad topics were general internal medicine (11.9%), public environmental occupational health (7.8%), pharmacology/pharmacy (6.3%), multidisciplinary sciences (5%), and infectious diseases (3.4%). About 67% were primary research articles, 16% were reviews, and the remaining were editorials letters (11.5%), meeting abstracts/proceedings papers (5%), and document corrections (0.4%). The University of Sharjah was the leading UAE-affiliated organization achieving 26.3% of the publications and funding 1.8% of the total 1008 published research. This study features the research trends in COVID-19 research affiliated with the UAE and shows the future directions. There was an observable nationally and international collaboration of the UAE-affiliated authors, particularly with researchers from the USA and England. This study highlights the need for in-depth systematic reviews addressing the specific COVID-19 research-related questions and studied populations.

## 1. Introduction

The severe acute respiratory syndrome coronavirus 2 (SARS-CoV-2) that causes Coronavirus disease 2019 (COVID-19) pandemic emerged in China in December 2019 [1]. As of 10 April 2022, over 496 million confirmed cases and over 6 million deaths had been reported globally [2]. This global spread has also resulted in the emergence of several new SARS-CoV-2 variants and sub-variants with varied transmissibility, infectivity, pathogenicity, and fatality rates [3,4,5]. Parallel to the implemented non-pharmaceutical control measures, including social distancing, wearing face masks, and lockdowns, the COVID-19 pandemic has ignited scientific research in an unprecedented tandem. Scientific research aimed to understand and control the pandemic and its consequences, including, but not limited to, epidemiological, diagnostic, and therapeutic studies. Evidence-based findings on understanding the country-based epidemiology of the disease and the effectiveness of specific implemented control measures and interventions are paramount. This need for scientific research has also contributed to identifying and challenging the country’s capacities to carry out basic and advanced scientific research that addresses local research questions. 

In the United Arab Emirates (UAE), the first case of COVID-19 was detected on 29 January 2020 [6]. By 22 April 2022, there have been 896,273 cases and 2302 deaths registered in the country [7]. To contain this pandemic, the country saved no effort and resources. The UAE is one of the top countries to implement several response measures and vaccinate their population after the emergency authorization for the use of the vaccine. The mass testing initiative was one of the main implemented control measures. The UAE has adopted a nationwide mass screening and testing strategy for early identification and isolating positive cases and applying quarantine on their contacts [8,9]. This mass screening and testing policy were classified as “Open Public Testing” by the Oxford COVID-19 Government Response Tracker (OxCGRT) [10]. As of 22 April 2022, the UAE has conducted over 153 million (1,552,554.1 per 100,000 population) polymerase chain reaction (PCR) tests for SARS-CoV-2 [7]. The Sinopharm vaccine was the first vaccine to be authorized for use in the UAE and it was, in addition to all other vaccines, given free of cost to all residents of the UAE [11]. Later, other types of vaccines were approved and used in the country. By 22 April 2022, 24,630,670 vaccine doses were given (249.04 per 100 people). The percentage of the eligible population who received one dose of the COVID-19 vaccine is 100.0%, whereas the percentage of the eligible population fully vaccinated against COVID-19 was 97.5% [7]. These achievements have placed the UAE in the top rankings for the highest rate of mass testing and vaccination.

Scientific research is an integral part of the fight against the COVID-19 pandemic and in addressing local research problems. Scientific research is very necessary, particularly when the country’s scientific capacity and infrastructure allow for that. The UAE is home to a wide range of universities and research centers, both public (three universities) and private (eight universities) [12], that host high-caliber researchers from all over the world. In the past two decades, the UAE has witnessed strong growth in scientific research output [13]. In the context of research on COVID-19, several studies documenting the local efforts in fighting the pandemic and in addressing several research questions were produced from the UAE. Nevertheless, analyzing the COVID-19-related research productivity from researchers affiliated with the UAE is necessary to inform the contribution of the UAE not only in the fight against the pandemic but also in creating scientific knowledge and enriching the literature that contributes to containing the pandemic and understating its consequences. 

Although the world has started to recover, in many countries, the COVID-19 pandemic is still ongoing. In other countries, including the UAE, the pandemic is being managed due to the high vaccination coverage and other non-pharmaceutical interventions. The emerging viral variants and subvariants are taking more lives and imposing extra challenges on healthcare systems and countries’ economies. Fighting the pandemic through implementing pharmaceutical and non-pharmaceutical interventions is very critical. The country’s contribution to scientific research assessing several aspects and impact of the pandemic, effectiveness of the implemented interventions, and innovative future interventions to fight pandemics is very critical; therefore, it is important to evaluate the contribution of the UAE and key developments in COVID-19 research. Recently, bibliometric analysis attracted the attention of researchers in several fields, including medical and health sciences [14]. Furthermore, bibliometric analysis is a powerful tool to investigate research progress and achievements in terms of key areas, trends, influential studies, productive authors, institutes, and countries in any research field [15]. This systematic bibliometric analysis describes the research productivity on COVID-19 that has been produced during the first two years of the pandemic by researchers affiliated with the UAE organizations. 

## 2. Materials and Methods

### 2.1. Literature Search Strategy 

The Web of Science Core Collection (WoSCC) database was utilized to retrieve indexed publications related to COVID-19 published from the UAE. The WoSCC is a comprehensive database that includes all editions; Science Citation Index Expanded (SCI-EXPANDED), Emerging Sources Citation Index (ESCI), Social Sciences Citation Index (SSCI), Conference Proceedings Citation Index—Science (CPCI-S), Arts & Humanities Citation Index (A&HCI), and Conference Proceedings Citation Index-Social Science & Humanities (CPCI-SSH). The WoSCC database is the most frequently used database for bibliometric studies [15,16,17]. 

The database search was restricted to the period from December 2019 until December 2021, and the English language. COVID-19 concept key terms were used in the title search field. The second concept was related to the UAE/organizations/universities and the search field was limited to the affiliations section. There was no restriction on the type of documents that were included. The Preferred Reporting Items for Systematic reviews and Meta-Analyses (PRISMA) guidelines [18] that could be adapted and applied to bibliometric analysis were followed. The exact PRISMA guidelines that we followed in this bibliometric analysis were the database search and assessing studies’ eligibility.

The following searching keywords/terms were used; “Coronavirus” OR “Novel coronavirus” OR “Coronavirus disease 2019” OR “COVID-19” OR “COVID” OR “Severe acute respiratory syndrome 2” OR “SARS-CoV-2” (Title) and “United Arab Emirates” OR “U Arab Emirates” OR “UAE” OR “Emirates” OR “Abu Dhabi” OR “Ajman” OR “Dubai” OR “Fujairah” OR “Ras al-Khaimah” OR “Sharjah” OR “Umm al-Quwain” OR “Al Ain” (Address) not “Middle East Respiratory Syndrome” OR “MERS” OR “SARS” (Title) not “Pak Emirates Military Hospital” OR “PEMH” (Address) and 2004 or 2013 or 2016 or 2017 or 2018 (Exclude—Publication Years).

### 2.2. Studies Selection and Data Extraction 

Two reviewers independently screened the retrieved citations against the inclusion criteria (COVID-19-related research and UAE-affiliated author) and performed the study selection. Disagreements between reviewers were settled by discussion or a third reviewer if required. The data were downloaded in a comma-separated value and plain text format. The following characteristics were extracted from the included publications; publication title, listed authors name, journal name, year of publication, keywords, institution, funding agency/organization, country of origin, and citations count. The impact factor (IF) and quartile ranking of the journals were obtained from the *Journal Citation Reports* for the year 2020 released in June 2021 by Clarivate Analytics. 

### 2.3. Data Analysis

Several bibliometric indicators were analyzed, including the most prolific authors, leading organizations, leading journals, most frequently used keywords, etc. The calculated values were presented in frequencies and percentages. Furthermore, the data were imported into RStudio and VOSviewer software for further analysis. Bibliometrics: An R-tool freely available package was utilized to construct WordCloud (author keywords), thematic map, and topic dendrogram [19]. The VOSviewer software version 1.6.17 for MacOS was used to construct the co-authorship authors, co-authorship organizations, and co-authorship countries network visualization [20]. 

## 3. Results

The initial search yielded 1113 documents. After removing duplicates and retrieving citations from publications that did not meet the inclusion criteria, 1008 UAE-affiliated publications published in the English language were included in the final bibliometric analysis. The PRISMA flow chart for study selection is shown in Figure 1. In total, 828 publications were available to open access and 103 were early access publications; 298 publications were published in the year 2020 and 700 papers were published in 2021 (up to 28 December); however, 10 publications were early access publications published in 2021; thus, the total number of publications in 2021 was considered 710.

### 3.1. Web of Science Categories, Leading Funding Agencies, and Publishers

Table 1 shows the main characteristics of the included UAE-affiliated publications. The majority of the COVID-19 and UAE-affiliated publications were published as an original article. The top five most studied WoS categories were medicine general internal (*n* = 120), public environmental occupational health (*n* = 79), pharmacology pharmacy (*n* = 64), multidisciplinary sciences (*n* = 51), and infectious diseases (*n* = 34). The top leading funding agencies were the University of Sharjah (*n* = 18), National Institutes of Health-United States of America (USA) (*n* = 14), United States Department of Health Human Services (*n* = 14), European Commission (*n* = 10), King Saud University (*n* = 10), United Arab Emirates University (*n* = 9), Khalifa University of Science and Technology (*n* = 8), and New York University Abu Dhabi (*n* = 8). The top five leading publishers were Elsevier (*n* = 173), Springer Nature (*n* = 107), MDPI (*n* = 65), Wiley (*n* = 64), Frontiers Media SA (*n* = 58), and Taylor & Francis (*n* = 58). 

### 3.2. Top 10 Most Cited Publications 

The top 10 most cited UAE-affiliated publications are presented in Table 2. The majority (70%) of these 10 cited publications were published as original articles (*n* = 7). The most cited paper was “Surviving Sepsis Campaign: Guidelines on the Management of Critically Ill Adults with Coronavirus Disease 2019 (COVID-19)” published in two journals (*Intensive Care Medicine*, IF: 17.44, and *Critical Care Medicine*, IF: 7.5) received a total of 938 citations up to the searching date of the current study. This article discusses the recommendations to help healthcare workers/medical practitioners caring for critically ill patients infected with COVID-19. 

### 3.3. Trend and Shipping of Research 

To understand the research trend and shipping, we calculated the centrality and density measures based on KeyWords Plus (see Figure 2). The number of words was 250, and the minimum cluster frequency (per thousand documents) was selected at 5. The centrality demonstrates the strength of association between the plotted keywords in one cluster with another cluster, while the density represents the aggregate strength of the relationship/association between the plotted keywords in the same cluster [32]. The thematic map is divided into four themes/quadrants; (a) Motor Themes (top right) represents the bridge between other topics, (b) the Niche Themes (top left) represent highly developed topics, (c) the Basic Themes (bottom right) represent the basic and transversal topics currently under development, (d) while the Emerging or Declining Themes (bottom left) represent the emerging topics. Figure 2 shows that the topics are mainly plotted within Basic Themes and then shifted to Niche Themes. The topics in Basic Themes are mainly about the coronavirus disease/infection, outbreak, disease impact, and health management, whereas the topics in Niche Themes focus on the risk of transmission and infection in healthcare workers. 

Furthermore, based on factorial analysis, the topic dendrogram was generated. The method applied was multiple correspondence analysis, whereas the field was KeyWords Plus and selected at 20. The topics based on KeyWords Plus were grouped into two clusters, as shown in Figure 3. The first cluster (blue color) is grouped by stress, anxiety, and depression. In cluster 2 (red color), the disease (COVID-19) is coupled with SARS (SARS-CoV-2, the causative agent of COVID-19), and Wuhan (the place where the first case of COVID-19 was reported in late December 2019). The next part of cluster 2 is formed by Angiotensin-converting enzyme-2 (ACE2) and remaining all other keywords that contribute to disease transmission, infection, outcomes, impact on health, risk factors, epidemic and outbreak, and care (prevention). 

### 3.4. Leading Journals 

The leading journal was *PloS One* (*n* = 23), followed by *Dubai Medical Journal* (*n* = 21), *Medicine* (*n* = 19), *Frontiers in Public Health* (*n* = 16), and *International Journal of Environmental Research and Public Health* (*n* = 14), as shown in Table 3. The leading journals IF ranged from 1.889 (*Medicine*) to 3.709 (*International Journal of Environmental Research and Public Health*). 

### 3.5. Top 10 Leading Institutions

Of the 1008 UAE-affiliated publications, 26.3% (*n* = 265) affiliated with the University of Sharjah, followed by 16.9% (*n* = 170) affiliated with the United Arab Emirates University, and 7.5% (*n* = 76) affiliated with the Zayed University. The number of publications produced by the top 10 institutions ranged from 29 to 265 publications (Table 4).

### 3.6. Co-Authorship Authors

The minimum number of documents of an author was selected at five. Of the total involved authors (*n* = 6749) in the 1008 publications, 80 authors met the criteria and were plotted based on the total link strength (TLS). The authors with zero TLS were excluded; after the exclusion, only 29 authors were plotted. A total of four clusters were formed, and each color represents a different cluster. Cluster 1 consists of 10 items (red color), cluster 2 (8 items, green color), cluster 3 (8 items, blue color), and cluster 4 (3 items, yellow color). The most influential author was Rabih Halwani (see Figure 4).

### 3.7. Co-Authorship Organizations

The minimum number of publications of an organization was selected at five. Of the total involved organizations (*n* = 1869), only 125 organizations met the criteria and were plotted for network visualization. A total of 11 clusters were formed; the maximum cluster size was 22, while the minimum cluster size was three. Each color represents a different cluster. Based on the TLS, the top three most prolific organizations were the University of Sharjah, United Arab Emirates University, and King Saud University (see Figure 5).

### 3.8. Co-Authorship Countries

The minimum number of publications of a country was fixed at five. Of the total involved countries (*n* = 111), 70 countries met the criteria and were plotted for network visualization, as shown in Figure 3. A total of five clusters were formed. The maximum cluster size was 24 (cluster 1, red color), followed by cluster 2 (21 items, green color), cluster 3 (15 items, blue color), cluster 4 (7 items, yellow color), and cluster 5 (3 items, purple color). Each color represents a different cluster. Based on the TLS, the top three most collaborative countries were the United Arab Emirates, the USA, and England. The countries’ collaborations are presented in Figure 6.

### 3.9. Author Keywords WordCloud Map

The closely connected keywords were colored the same, which represents that these keywords might share something in common. The top 10 most widely used author keywords were COVID-19 (*n* = 551), SARS-CoV-2 (*n* = 106), coronavirus (*n* = 99), pandemic (*n* = 70), COVID-19 pandemic (*n* = 30), UAE (*n* = 26), United Arab Emirates (*n* = 25), mental health (*n* = 19), anxiety (*n* = 16), depression (*n* = 15), and public health (*n* = 15). The WordCloud map of author keywords is presented in Figure 7.

## 4. Discussion

The findings of this systematic bibliometric analysis document the significant contribution of researchers affiliated with UAE in COVID-19-related research during the first two years of the pandemic.

A previously published bibliometric analysis in March 2021 showed that there were 143,975 COVID-19-related publications in the Scopus search engine and only 6131 (4.26%) of these were collected from the Arab countries [33]. At that stage of the pandemic, the UAE (*n* = 719, 11.73%) was ranked the third Arab country in the number of COVID-19 publications after Saudi Arabia (*n* = 2186, 35.65%) and Egypt (*n* = 1281, 20.78%) [33]. In December 2021, our bibliometric analysis included 1008 publications related to COVID-19 research affiliated with at least one author from the UAE and indexed in the WOSCC database. This indicates a significant increase in the number of publications within approximately nine months; however, this research productivity is not comparable, as the present and the other [33] bibliometric analyses used different databases, and these countries were hit by the pandemic at different times and levels of severity.

The top five journals in publishing COVID-19-related papers by the UAE-affiliated authors published 93 papers (8.6% of the total publications), ranging from the emerging source of citations (Dubai Medical Journal) to IF of 3.39 (International Journal of Environmental Research and Public Health) in the top five journals. Based on the TLS analysis, the highest research collaboration for UAE researchers in terms of countries was within the UAE, then with the USA and England. This is consistent with other bibliometric analyses showing that the USA is the world-leading country in terms of COVID-19 research contribution [33,34]. In our analysis, this was more apparent in the TLS analysis at an organizational and authors’ level. The majority of UAE authors seem to collaborate with colleagues from their organizations or other organizations within the UAE. This is a very similar trend to a visualization mapping of COVID-19 in Southeast Asia, which showed that research collaboration within each country, such as Malaysia, Singapore, and Indonesia, more than international research collaboration [35]. Although it is a natural trend for researchers to collaborate with colleagues from their organizations and geographical region, it is important to highlight that international research collaboration should be expanded beyond the geographical borders, especially in pandemics circumstances. 

Our analysis of the published document types showed that over 67% of the published COVID-19 publications affiliated with the UAE were articles. This indeed indicates that UAE scientists were focusing on being involved in primary research during the first two years of the pandemic, as well as the country’s contribution in funding and facilitating conducting original studies. A similar bibliometric analysis conducted in Africa showed that 46.7% of the publications were primary research papers [36]. Furthermore, the highest cited article in our analysis received 938 citations within less than two years, indicating the considerable scientific value that it adds to the COVID-19 literature; however, it is difficult to compare research productivity across countries at this early stage because the scientific output is still evolving and days, weeks, and months are making a significant difference in productivity output. For example, Guleid and colleagues [36] conducted their search based on the first year of the pandemic, while this research focuses on the first two years. 

Our findings show that the University of Sharjah is the leading institution in the UAE in publishing COVID-19-related research. For sure, this high productivity compared to other UAE Universities is not due to university age because the University of Sharjah was established in 1997 and produced nearly 36% more publications than United Arab Emirates University, which was established in 1976. It has been reported in the literature that funding is one of the main determinants of scientific activities [37]. In this bibliometric analysis, it appears that one of the factors contributing to this productivity is the research fund; The University of Sharjah funded 18 research projects compared to 9 projects funded by the United Arab Emirates University. Another contributing factor to high research productivity is collaboration, especially cross-sector and cross-discipline collaborations [38]. The University of Sharjah was the top UAE organization in the strength of organizational collaboration, as shown by the co-authorship organizations network visualization based on TLS.

To the best of our knowledge, this is the first bibliometric analysis to describe research on COVID-19 affiliated with the UAE. The documents included in this study were only limited to the WoSCC; however, WoSCC is one of the world’s largest citation databases. Furthermore, the search utilized all databases of the WoS and implemented a systematic selection and reviewing strategy. Based on WoS categories, our analysis showed that the top five most studied were medicine general internal, public environmental occupational health, pharmacology pharmacy, multidisciplinary sciences, and infectious diseases, respectively. These areas are general specialty areas, which makes it difficult to understand the research gaps in each category. The nature of this type of bibliometric analysis did not allow for a systematic method of data extraction, as this would limit the visualization presentation of the results; therefore, the process of selection of the included document had to be conducted manually by two reviewers. The results of this bibliometric analysis must be interpreted in line with the methodology limitation. For example, the number of publications per organization or author does not indicate the research impact and implications; however, it shows the research trends in this area and identifies gaps in research for funders, organizations, and researchers to draw plans for future research. Moreover, the results of this bibliometric analysis do not cover pre-prints as well as yet-to-be-indexed studies reported by researchers affiliated with the UAE.

## 5. Conclusions

This study captured the relevant literature in COVID-19 research affiliated with the UAE and showed the research trends and future directions. Although the majority of authors affiliated with the UAE collaborated with colleagues from the UAE, they have contributed considerably to the global COVID-19 research productivity by achieving a wide range of international collaborations, including the USA and England. Research collaborations and funding seem to be the main drivers of increasing research productivity for COVID-19 research in the UAE. This study is limited to one country, uses one database search, and identifies the most studied areas as broad topics, including general medicine, public health, pharmacology, multidisciplinary sciences, and infectious diseases; therefore, this bibliometric analysis advocate for in-depth systematic reviews highlighting the addressed COVID-19 research-related questions and studied populations to identify gaps in evidence to direct future research.

## Figures and Tables

**Figure 1 ijerph-19-07753-f001:**
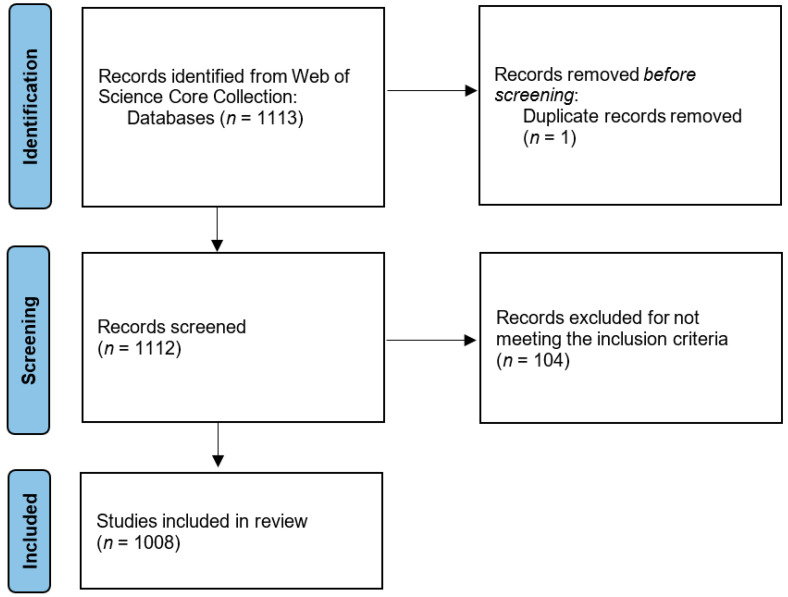
PRISMA Flow Diagram.

**Figure 2 ijerph-19-07753-f002:**
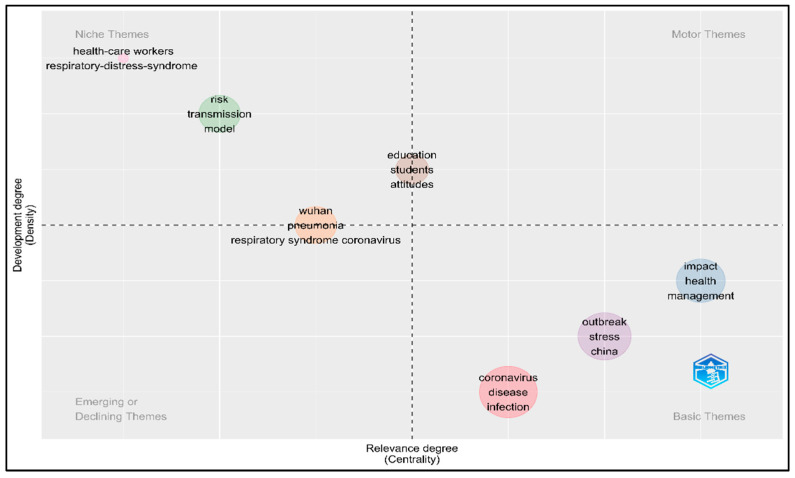
Thematic map based on KeyWords Plus.

**Figure 3 ijerph-19-07753-f003:**
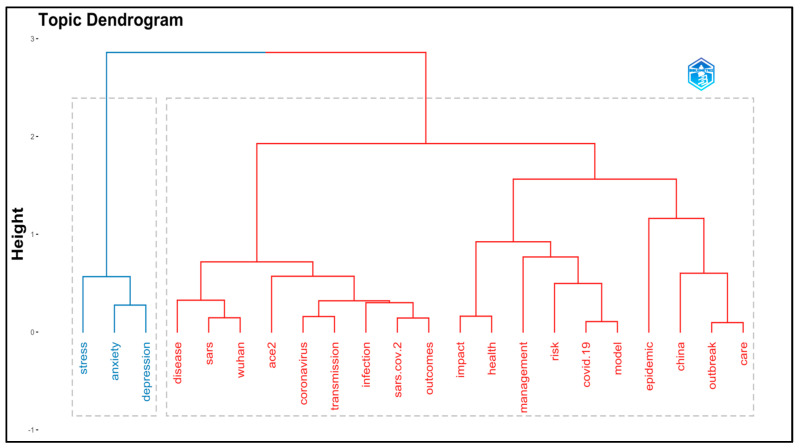
Topic dendrogram based on KeyWords Plus.

**Figure 4 ijerph-19-07753-f004:**
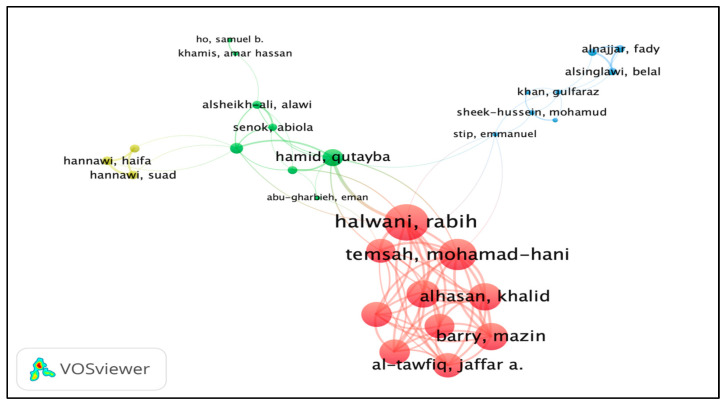
Co-authorship authors network visualization based on TLS. Different colors indicate different clusters, while the nodes indicate the contribution of an author, and the links represent the linkage/collaboration between authors. The bigger the node, the higher contribution of an author; the thicker the linkage/line, the stronger the collaboration between authors.

**Figure 5 ijerph-19-07753-f005:**
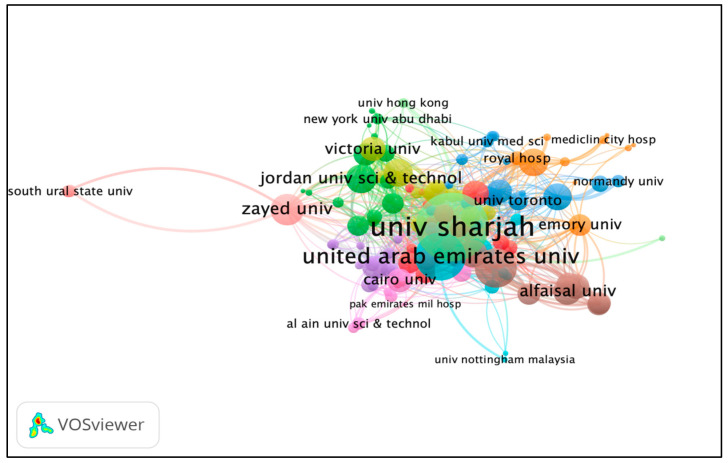
Co-authorship organizations network visualization based on TLS. Different colors represent different clusters, while the nodes represent the contribution of an organization, and the links represent the linkage/collaboration between organizations. The bigger the node, the higher contribution of an organization; the thicker the linkage/line, the stronger the collaboration between organizations.

**Figure 6 ijerph-19-07753-f006:**
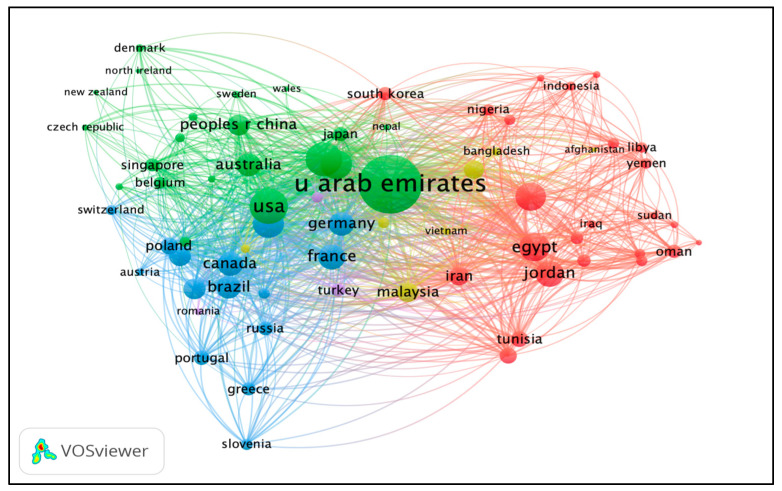
Co-authorship countries’ network visualization based on TLS. Different colors represent different clusters, while the nodes represent the contribution of a country while the links represent the linkage/collaboration between countries. The bigger the node, the higher contribution; the thicker the linkage/line, the stronger the collaboration.

**Figure 7 ijerph-19-07753-f007:**
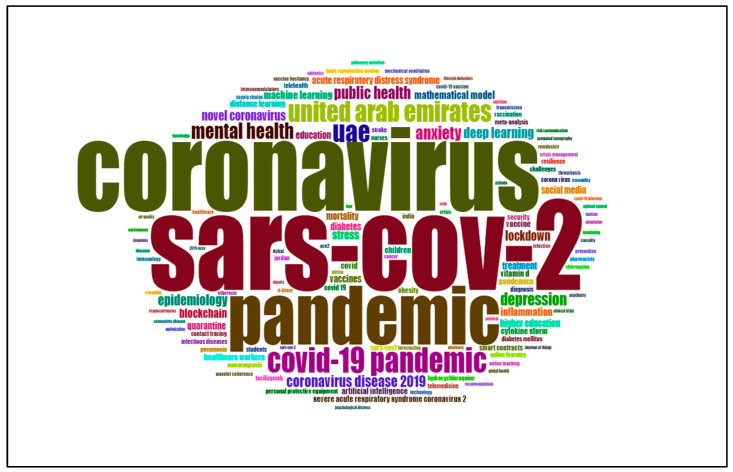
The author keywords WordCloud mapping.

**Table 1 ijerph-19-07753-t001:** Main characteristics of the included UAE-affiliated publications.

Description	Results
Main information about data	
Time span	2020–2021
Sources (Journal, books, magazines, etc.)	594
Number of documents	1008
Average citations per document	9
Average citations per year per document	6
Document types	
Articles	676
Reviews	162
Editorials/Letters	116
Meeting abstracts/Proceedings paper	50
Document corrections	4
Authors	
Single-authored documents	60
Multi-authored documents	948
Total number of authors in the 1008 publications	6749
Documents per Author	0.149
Authors per document	6.71

**Table 2 ijerph-19-07753-t002:** Top 10 most cited publications published by UAE-affiliated authors.

Rank	Paper	Publication Type	Citations	Citations per Year
1=	* Alhazzani, et al. (2020) [21]	Article	938	469
1=	* Alhazzani, et al. (2020) [22]	Article	938	469
2	Ammar, et al. (2020) [23]	Article	507	253.5
3	Al-Shamsi, et al. (2020) [24]	Review	299	149.5
4	Fan, et al. (2020) [25]	Article	150	75
5	Zaremba, et al. (2020) [26]	Article	132	66
6	Aziz, et al. (2020) [27]	Review	103	51.5
7	Bruinen de Bruin, et al. (2020) [28]	Article	98	49
8	Ammar, et al. (2020) [29]	Article	97	48.5
9	Ibn-Mohammed, et al. (2021) [30]	Article	93	93
10	Hasan, et al. (2021) [31]	Review	87	87

* Article is sponsored by two different societies and each society published the guidelines in their journal. Society of Critical Care Medicine—Journal: *Critical Care Medicine* [22] and European Society of Intensive Care Medicine—Journal: *Intensive Care Medicine* [21]. Both publications are presented for transparency and only one of them was accounted for in the analysis to avoid double counting and over-estimation.

**Table 3 ijerph-19-07753-t003:** Top five leading journals publishing the highest number of COVID-19-related publications by the UAE-affiliated authors.

Rank	Journals	IF (Five Year IF)	Quartile Ranking (Category Rank)	Publications	Publisher Address
1	*Plos One*	3.24 (3.788)	Q2 (26/72)	23	Public Library Science1160 Battery Street, STE 100, San Francisco, CA 94111
2	*Dubai Medical Journal*	-	-	21	Karger, Allschwilerstrasse 10, CH-4009 Basel, Switzerland
3	*Medicine*	1.889 (2.351)	Q3 (99/167)	19	Lippincott Williams & Wilkinstwo Commerce SQ, 2001 Market ST, Philadelphia, PA 19103
4	*Frontiers in Public Health*	3.709 (4.022)	Q1, SSCI edition (36/176), Q2, SCIE edition (62/203)	16	Frontiers Media Saavenue Du Tribunal Federal 34, Lausanne CH-1015, Switzerland
5	*International Journal of Environmental Research and Public Health*	3.39 (3.789)	Q1, SSCI edition (42/176), Q2, SCIE edition (68/203)	14	MDPI, ST Alban-Anlage 66, CH-4052 Basel, Switzerland

Note: The IF and quartile ranking of the journals were obtained from the Journal Citation Reports, 2020.

**Table 4 ijerph-19-07753-t004:** Top 10 leading UAE institutions publishing COVID-19-related publications.

Rank	Institution	Number of Documents ^1^
1	University of Sharjah	265
2	United Arab Emirates University	170
3	Zayed University	76
4	Ajman University	67
5	Mohammed Bin Rashid University	60
6	Khalifa University for Science and Technology ^2^	59
6	Al Ain University	59
8	Dubai Hospital	35
9	Rashid Hospital	32
10	New York University Abu Dhabi	29

^1^ Numbers allow for overlapping as some publications might be produced from more than one UAE-affiliated institution. ^2^ Khalifa University for Science and Technology appeared with this name and Khalifa University. Both are referring to the same University; therefore, they were combined.

## Data Availability

Data are available upon reasonable request.

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
