# Peer review of "SARS-CoV-2 and COVID-19 Research Trend during the First Two Years of the Pandemic in the United Arab Emirates: A PRISMA-Compliant Bibliometric Analysis"

_ijerph, 2022, doi:10.3390/ijerph19137753_

Round 1

Reviewer 1 Report

This paper evaluates 'output' from the UEA related to COVID-19. I have the following suggestions/comments:

1. Lines 45-46: "Scientific research aimed to understand and control the pandemic and its consequences including." is missing context; including what?

2. Table 2: "*Article is published in 2 different journals as a protocol and guidelines". Is it appropriate to publish the same paper in 2 different journals? Was this done at the request of the journals?

3. Discussion; lines 290-291: "Although the world started to recover, in several countries the COVID-19 pandemic is still ongoing." I suspect the COVID-19 pandemic is still ongoing in many countries, although potentially more manageable given high vaccination rates.

Author Response

Response to Reviewer 1

Comment: This paper evaluates 'output' from the UAE related to COVID-19. I have the following suggestions/comments:

Response: The authors thanks the reviewers for their time and efforts on evaluating our work and for providing constructive comments and suggestions. The authors have now provided a point-by-point response to the suggestions/comments raised.

Comment 1. Lines 45-46: "Scientific research aimed to understand and control the pandemic and its consequences including." is missing context; including what?

Response: We thank the reviewer for spotting the missing context. The authors have elaborated more on the missing context. (Please refer to lines 46-47)

Comment 2. Table 2: "*Article is published in 2 different journals as a protocol and guidelines". Is it appropriate to publish the same paper in 2 different journals? Was this done at the request of the journals?

Response: We thank the reviewer for raising this important observation. In fact, this article is sponsored by two different societies and each society published the guidelines in their journal. Society of Critical Care Medicine published in Critical Care Medicine and the European Society of Intensive Care Medicine published Intensive Care Medicine. We included both publications for transparency. However, we have only counted one in the analysis to avoid double counting and over-estimation. Again, we thank the reviewer for highlighting this important point; the authors explained this in the footnote of Table 2 to make it clear to the reader. (Please refer to lines 190-194)

Comment 3. Discussion; lines 290-291: "Although the world started to recover, in several countries the COVID-19 pandemic is still ongoing." I suspect the COVID-19 pandemic is still ongoing in many countries, although potentially more manageable given high vaccination rates.

Response: We thank the reviewer for their comment. The authors have now modified the sentence as per the reviewer’s suggestion. Based on reviewer 2 suggestion, this paragraph is moved to the introduction. (Please refer to lines 85-89).

In the revised manuscript, all changes are provided in blue color.

Reviewer 2 Report

Dear Authors, 

in my opinion , your research is interesting but it must be improved in some sections. 

1) In the introduction, you must highlight the gap that your research wants to fill (in this version you have written it in the discussion section). Moreover it's important to add the structure of the paper.

2) The material and method section is clear and well written.

3) The results are adequate.

4) In the discussion section I suggest to deepen your analysis. You need to better underline all the results. It could be important to add theoretical and practical implications of your research.

5) In the conclusion you must add some limitations of the research.

Author Response

Response to Reviewer 2

Comment: in my opinion, your research is interesting but it must be improved in some sections. 

Response: The authors thanks the reviewer for their time and efforts in evaluating our work and providing constructive comments and suggestions. The authors have now provided a point-by-point response to the comments and suggestions raised.

Comment 1) In the introduction, you must highlight the gap that your research wants to fill (in this version you have written it in the discussion section). Moreover, it's important to add structure to the paper.

Response: We thank the reviewer for highlighting this point and we agree regarding the importance of the structure of the paper. The authors moved the text highlighting the gap that this research fills to the introduction section and added text to clarify the context of the paper. (Please refer to lines 85-100)

Comment 2) The material and method section is clear and well written.

Response: We thank the reviewer for this pleasing comment. We are happy to hear that the material and method sections are clear and well written.

Comment 3) The results are adequate.

Response: We thank the reviewer for this pleasing comment. We are happy to hear that the results are adequate.

Comment 4) In the discussion section I suggest deepening your analysis. You need to better underline all the results. It could be important to add theoretical and practical implications to your research.

Response: We thank the reviewer for this pleasing comment. Generally, in bibliometric analysis, the discussion focuses only on the main results. However, based on the reviewer’s comments we provided a further discussion of the results (please refer to lines 310-312, 324-327, and 353-365). The authors have also added the conclusion from these additions to the conclusion section (please refer to lines 389-393).

Comment 5) In the conclusion you must add some limitations of the research.

Response: We thank the reviewer for their comments. The authors expanded more on the limitations section (please refer to lines 382-383) and added limitations to the conclusion (please refer to lines 386-388).

In the attached revised manuscript, all changes are provided in blue color.

Round 2

Reviewer 2 Report

Dear Authors, 

I appreciate your efforts to improve the paper. In my opinion, it can be published in the present form. 

Good luck